# Nitrogen Fertiliser Immobilisation and Uptake in the Rhizospheres of Wheat and Canola

Ben A. Rigby [1,2,*], Niloufar Nasrollahi [1], Corinne Celestina [1,2], James R. Hunt [1,2], John A. Kirkegaard [3] and Caixian Tang [1]

1   Department of Animal, Plant and Soil Sciences, AgriBio Centre for AgriBiosciences, La Trobe University, Bundoora, VIC 3086, Australia; N.Nasrollahi@latrobe.edu.au (N.N.); C.Celestina@latrobe.edu.au (C.C.); J.Hunt@latrobe.edu.au (J.R.H.); C.Tang@latrobe.edu.au (C.T.)
2   School of Agriculture and Food, Faculty of Veterinary and Agricultural Sciences, University of Melbourne, Melbourne, VIC 3052, Australia
3   CSIRO Agriculture and Food, GPO Box 1700, Canberra, ACT 2601, Australia; john.kirkegaard@csiro.au
*   Correspondence: barigby@student.unimelb.edu.au

**Abstract:** Immobilisation of fertiliser nitrogen (N) by soil microorganisms can reduce N availability to crops, decreasing growth and yield. To date, few studies have focussed on the effect of different plant species on immobilisation of fertiliser N. Canola (*Brassica napus*) is known to influence the soil microbiome and increase mineral N in soil for future crops compared with cereals. We tested the hypothesis that canola can reduce immobilisation of fertiliser N by influencing the composition of the rhizosphere microbiome. To investigate this, we conducted a glasshouse soil column experiment comparing N fertiliser uptake between canola and wheat (*Triticum aestivium*) and partitioning of fertiliser N between plants and microorganisms. Plants were grown in soil to which high C:N ratio wheat residues and $^{15}N$-labelled urea fertiliser were applied. There was no difference between wheat and canola in fertiliser N uptake despite differences in fungal community composition and the carbon metabolising enzyme alpha-glucosidase in the rhizosphere. Canola obtained more soil-derived N than wheat. There was no significant difference in the rhizosphere bacterial communities present between wheat and canola and unplanted controls. Our results highlight the capacity of canola to increase mineralisation of soil N compared with wheat although the study could not describe the microbial community which facilitated this increase.

**Keywords:** canola; nitrogen immobilisation; nitrogen mineralization; microbial ecology; wheat

## 1. Introduction

Nitrogen (N) deficiency is considered the main impediment to achieving water-limited potential yield in Australian wheat (*Triticum aestivum*) production systems [1]. Farmers are often conservative with fertiliser application rates due to climate variability and other risk factors which influence yield [2]. In addition to inadequate application rates, Australian wheat crops are estimated to use only 40% of N fertiliser in the season it is applied [3]. Identifying ways to increase crop N-use efficiency (NUE) provides the dual benefit of reducing the rate required for fertiliser application while reducing the risk of potentially damaging N loss to the environment.

A recent simulation study projected that microbial immobilisation was the main source of N fertiliser inefficiency in southern Australian cropping systems with retained stubble [4]. Widespread adoption of no-till cropping in Australian cereal systems [5] has resulted in the retention of significant amounts of C-rich crop residues in cropped paddocks. Rates of immobilisation are particularly high in grain production systems where residues are retained because the presence of high C:N ratio cereal crop residues encourages microbes to use available N to support residue decomposition [6].

Uptake of soil nutrients by microorganisms is essential for nutrient cycling into plant available forms [7]. However, immobilisation reduces the availability of mineral N, an important N source for plants. If fertiliser is applied at sowing, this early competition for N may reduce the vigour of young crops potentially affecting yield. Fertiliser applications that coincide with times of high crop demand (i.e., stem elongation in wheat) can increase the uptake of applied N by plants. However, the abundance of C-rich crop residues ensures that some fertiliser is still immobilised. Nitrogen partitioning in the soil–plant system between microorganisms and plants is thought to change over time, with microbes accumulating more in the short term and plants more in the long term [8]. The spatiotemporal context of the competition for resources and the influence of environmental factors upon N transformations results in a highly dynamic interaction between plants and microorganisms. To date, few studies have investigated the effect of plant species upon microbial immobilisation directly.

Canola (*Brassica napus*) is a common break crop grown in rotation with wheat [9]. The break crop effect provided by canola to subsequent wheat crops was initially thought to be primarily due to suppression of root pathogens by glucosinolates, a secondary metabolite produced by *Brassica* species [10]. However, Kirkegaard et al. [11] demonstrated that mineral N accumulation following a canola crop was higher than following a cereal crop, and often equivalent to legume crops. Ryan et al. [12] attributed this increase in mineral N following canola to changes in the microbial community composition which emerged from differences in the chemical composition of plant residues. Studies have shown that canola root tissues can influence individual soil microorganisms such as $N_2$-fixing *Azospirillum* sp. [11] and the soil-borne pathogen *Gaeumannomyces graminis* [13]. They also appear to influence specific groups of soil microorganisms such as ammonia oxidisers [14]. O'Sullivan et al. [14] showed that wheat required less N fertiliser when grown after canola compared with after pasture, and that canola reduced nitrification and increased immobilisation and remobilisation rates of N in the rhizosphere compared with wheat. These effects of canola on rhizosphere biology may provide benefits to crop production if immobilised fertiliser N is mineralised and acquired by plants in season.

Immobilisation and mineralisation of N are opposing processes of the soil N cycle and determine the distribution of N between mineral and organic forms. Mineralisation converts soil organic matter (SOM) containing N into mineral N, predominantly $NH_4^+$. The size of the microbial biomass is often reported as the major factor determining N mineralisation [15]. Elsewhere, it is suggested that the microbial community composition may influence mineralisation of SOM [16]. Mineralisation and immobilisation are processes which are carried out by most of the microbial population rather than by specific phyla [17]. The nutritional demand of the microbial population is likely a key variable influencing both processes, especially when microorganisms are stimulated by the addition of external nutrients [18]. The universal requirement of organisms for N as a macronutrient ensures that in environments with low N availability the competition for available N will be high.

The focus of this study was to determine if canola and wheat differed in their ability to capture applied fertiliser N in the presence of C-rich crop residues, and whether this was related to a reduction in the immobilisation of N in canola rhizospheres as a result of induced changes in microbial community composition and activity. We hypothesised that reduced microbial biomass N (MBN) derived from fertiliser in the canola rhizosphere would enable canola to obtain more fertiliser N compared with wheat.

## 2. Materials and Methods

### 2.1. General

The experiment took place in a glasshouse at AgriBio, Bundoora, Victoria, Australia. The soil used throughout the experiment was collected from Normanville, Victoria, Australia (35°50′39″ S, 143°44′56″ E). The soil used was a Vertic Calcarosol [19] with a sandy clay loam texture, bulk density of 1.2 g/cm$^3$, and pH of 6.5 (CaCl$_2$ 1:5). Initial soil total N was 1.17 g/kg (±0.04) which is similar to other dryland, long-term cropping sites in



Australia [20]. Total mineral N ($NH_4^+ + NO_3^-$) of the soil was 22.1 ± 5.6 mg/kg. Approximately 500 kg of surface soil (0–15 cm) was collected in April 2018 from a paddock which had grown a wheat crop in the previous growing season. Wheat stubble with a C:N ratio of 70:1 was also collected from this site at the time of soil collection. Glasshouse temperature was 22 °C during daylight hours (14 h) and 14 °C at night (10 h). Light was supplemented using sodium halide lamps during the 14-h daylight period if external irradiance fell below 170 W/m$^2$.

## 2.2. Experimental Design

The experiment was a balanced two-way factorial randomised complete block design with four replicates, two crop species (wheat and canola), and two nitrogen fertilisation rates (low, 5 kg-N ha$^{-1}$ equivalent and moderate, 61 kg-N ha$^{-1}$ equivalent commercial rate for Australian dryland grain cropping systems). This design was replicated for both harvest times (anthesis and maturity). Unplanted control treatments were given the same N rates; however, they were not included in the balanced two-way factorial experimental design. Due to the herbicide history of the collection site, wheat (cv. Razor CL plus, Australian Grain Technologies) and canola (hybrid 44Y90 CL, Pioneer Seeds) cultivars with Clearfield® imidazolinone tolerance were selected to ensure that residual herbicide did not adversely influence plant growth.

Plants were grown in PVC columns containing 12.7 kg of air-dried soil sieved to ≤2 mm. Soil was wet and maintained at ~80% field capacity (20.5% *w/w*) during plant growth. The top 10 cm of soil had 18 g of ground (5 mm sieve) wheat stubble incorporated uniformly (equivalent to 10 t ha$^{-1}$). The incorporation of high-C wheat residue into surface soil with low mineral N (12 kg-N ha$^{-1}$ equivalent) was to encourage a competitive environment for fertiliser N uptake between plants and microorganisms. Columns were 60 cm tall to ensure roots were not confined to the upper 10 cm of columns where the stubble was added. Additionally, the top 10 cm received basal elemental nutrients (54 K, 49 Ca, 41 P, 33 S, 5 Mg, 5 Mn, 2 Zn, 1.5 Cu, 0.8 Fe, 0.1 Mo mg kg$^{-1}$) to ensure no limitations of other nutrients for plant growth. Nutrient solutions were mixed thoroughly through the top 10 cm before planting. Seeds were germinated on wet paper towel in a Petri dish and sown into the surface soil. Wheat plants were sown to a depth of 20 mm and canola at 15 mm. Unplanted controls received the same nutrient application and residue incorporation rates.

The two N fertiliser rates, low (10 mg N) and moderate (116 mg N) were applied to each of the crops. The N fertiliser used throughout was urea ($CH_4N_2O$), with 1% enrichment with $^{15}N$ to trace the fate of fertiliser N in plants and soil. The N fertiliser was first applied when three leaves had fully emerged on wheat plants, and the moderate-N treatment had two subsequent applications 14 days apart (38.6 mg at each application). The N fertiliser was applied to the soil surface with the water required to maintain moisture content. Plant and soil samples were taken when wheat reached anthesis (Z 65) [21] 57 days after sowing, and maturity-harvest ripe (Z89-92) 116 days after sowing. Canola plants were harvested at the same time as wheat and were not physiologically mature at the final harvest due to the indeterminate nature of canola. We estimate the canola to be at 30% pods ripe stage (83) [22]. Soil from the unplanted control was also sampled at the same time.

## 2.3. Plant Sampling and Analysis

Senesced leaves were collected during the growth period as they detached from the plants to prevent decomposition. At harvest, leaves, spikes (wheat), and pods (canola) were removed at the abscission layer between the respective tissue and the stem, and stems were then cut and removed at the soil surface. Spikes (wheat) and pods (canola) were analysed with immature grain in situ for the anthesis harvest, while the mature grain was threshed at maturity and analysed separately. All of the sampled tissues described above were included in the measurement of shoot N.

After sampling, all partitioned plant tissues were placed in an oven at 70 °C for 48 h. Each section of the shoot material was then weighed. All plant material was then ground and milled in preparation for nutrient analysis. The concentration of N in plant tissues from the anthesis sampling time was determined through a 2400 Series II CHNS/O Elemental Analyser (PerkinElmer, Waltham, MA, USA). Plant tissue and soil N and $^{15}$N% at the maturity sampling time were determined via mass spectrometry using an IRMS hydra 20-20 isotope ratio mass spectrometer (Sercon Limited, Cheshire, UK) combined with an ANCA-S/L sample preparation unit (Europa Scientific, Crewe, UK).

The percentage of applied $^{15}$N fertiliser uptake by plants (uptake, %) was calculated using the following formula [23]:

$$\text{Uptake (\%)} = 100 \times [(\text{TN})(c - 0.3663)]/[(\text{FN})(f - 0.3663)] \tag{1}$$

where, TN is the total amount of N in the plants (mg per column), c is the $^{15}$N% of the plant samples, 0.3663 is the assumed natural abundance of $^{15}$N, FN is the total amount of fertiliser N applied per column (mg), and f is the $^{15}$N% in the fertiliser. This value was then used to calculate fertiliser N uptake by plants in mg.

### 2.4. Soil Sampling and Analysis

After plants were harvested, rhizosphere soil was collected from the top 0–10 cm in each column. Loosely bound soil was shaken from the roots and tightly bound soil within 5 mm of the root surface was collected (rhizosphere soil). Bulk soil ($\geq$5 mm from root surface) from the 0–10 cm and 10–60 cm sections were sampled separately and all soil samples were sieved to <2 mm prior to analysis.

Soil pH was measured in 0.01 mol L$^{-1}$ CaCl$_2$ (1:5 ratio, soil:solution) after being shaken end-on-end for an hour and centrifuged (at 3000 rpm for 5 min) as per method 4B1 of Rayment and Lyons [24]. Moisture content of soil sections was determined at each harvest by weighing soil prior to and after 24 h oven drying at 105 °C. The values obtained were used to correct nutrient measurements on a mass basis of oven dry soil. All of the following soil results were calibrated to be shown as a mass basis of dry soil as per method 2A1 of Rayment and Lyons [24].

Analysis of soil nitrate and ammonium proceeded following extraction with 2 mol L$^{-1}$ KCl (shaking end-on-end for 1 h), centrifugation (3000 rpm for 5 min), and filtering supernatant through Whatman #42 filter paper (Whatman International, Maidstone, England). Extracts were stored at $-20$ °C before determination of extractable nitrate and ammonium using the flow-injection analyser via Quikchem 8500 Series II (Lachat Instruments, Loveland, CO, USA) system as detailed in the Quikchem manuals (12-10706-2-F and 12-107-04-1-B, respectively).

MBN in rhizosphere soil was determined by using a variation of the chloroform fumigation-extraction method as described by Vance et al. [25], within 24 h of sampling. Briefly, fumigated samples were incubated in a desiccator with approximately 50 mL of ethanol-free chloroform for 24 h at 25 °C in darkness. Fumigated and unfumigated soil samples were extracted using 0.05 mol L$^{-1}$ K$_2$SO$_4$ solution in a ratio of 1:5 soil to solution. A 0.05 mol L$^{-1}$ K$_2$SO$_4$ solution was used instead of 0.5 M K$_2$SO$_4$ for extraction due to the low $^{15}$N enrichment rate and to reduce salt content of freeze-dried samples to allow better detection of the isotope ratio. Samples were shaken end-over-end for 1 h followed by filtration through Whatman #42 filter paper. Following storage at $-20$ °C, extracts for MBN were freeze-dried and salt extracts were then analysed via IRMS hydra 20-20 isotope ratio mass spectrometer (Sercon Limited, Cheshire, UK) combined with an ANCA-S/L sample preparation unit (Europa Scientific, Crewe, UK). MBN is expressed as the difference between fumigated and unfumigated samples.

Extracellular enzyme activity of cellulose-decomposing enzymes in the rhizosphere soil was determined by high-throughput fluorometric measurement of 4-methylumbelliferone labelled substrates as outlined by Bell et al. [26]. Four enzymes were measured to assess the activity of the decomposer community present in each treatment to detect differences

between plant species. These were alpha-glucosidase (AG), beta-glucosidase (BG), cellobiase (CB), and beta-xylosidase (XYL). Soil was incubated at 25 °C for 3 h and fluorescence of microplates was detected using a CLARIOstar microplate reader (BMG LABTECH, Ortenburg, Germany).

DNA extraction and 16S rRNA and ITS diversity profiling were performed by the Australian Genome Research Facility on an Illumina MiSeq platform (San Diego, CA, USA). A 300-bp target was amplified from the V3–V4 region of the 16S rRNA gene using primers 341F (5′-CCTAY GGGRB GCASC AG) and 806R (5′-GACTA CNNGG GTATC TAAT) [27] and an approximately 230-bp target was amplified from the ITS1–ITS2 region of the internal transcribed spacer (ITS) using primers ITS1f (5′-CTT GGT CAT TTA GAG GAA GTA A) and ITS2 (5′-GCT GCG TTC TTC ATC GAT GC) [28,29].

Raw, demultiplexed fastq files were re-barcoded, joined, and quality filtered using the UPARSE pipeline [30]. Joined paired-end reads were quality-filtered by discarding reads with total expected errors >1 and removing singletons. Operational taxonomic units (OTUs) were clustered with a minimum cluster size >2 and 97% similarity cut off using the UPARSE-OTU greedy heuristic clustering algorithm. Taxonomic assignments were performed using the USEARCH UTAX algorithm with reference databases created using the RDP 16S (version 16) and UNITE ITS (version 7) training datasets (available at https://www.drive5.com/usearch/ accessed 8 November 2021). The minimum percentage identity required for an OTU to consider a database match a hit was 80%. All OTUs with a taxonomic confidence threshold less than 80% were denoted as 'unassigned'. The OTUs identified as chloroplasts and mitochondrial DNA were removed from the data set. A phylogenetic tree was constructed using the UPGMA algorithm in MUSCLE [31].

### 2.5. Statistical Analysis

Genstat version 19 (VSN International Limited, Hampstead, UK) was used for all statistical analyses of plant and soil data. Shapiro–Wilk test for normality was completed on each variate before analysis of variance (ANOVA) was performed and homogeneity of variance was assessed via plotting residuals and fitted values. Two-way analysis of variance (ANOVA) was used to assess differences in plant species and N treatment data collected at both sampling times. The results for the soil enzyme assay which included the controls as part of an unbalance experimental design were analysed using the unbalanced ANOVA function.

Bar charts of relative abundance of bacterial and fungal phyla were produced in R version 3.5.3 (R Core Team 2018) using the package phyloseq [32] with the assistance of ggplot2 [33] and RColorBrewer [34]. Before plotting, spurious reads were removed using a 0.005% relative abundance cut off [35]. Weighted and unweighted UniFrac distances [36] between samples were calculated from raw data, normalised to relative abundance, and then the relationships and differences between treatments were visualised with principal coordinates analysis (PCoA).

R package 'mvabund' [37] was used to test multivariate hypotheses about treatment effects and univariate hypotheses about species-by-species effects (i.e., the detection of differentially abundant OTUs). For this procedure, unrarefied sequence counts were modelled on negative binomial distributions in the generalized linear models and *p*-values were adjusted to control for the family-wise error rate.

## 3. Results

### 3.1. Differences in N Partitioning between Soil Microorganisms and Plant Shoots

There was no difference in the total MBN in the wheat and canola rhizosphere soils (Figure 1). Fertiliser N made up more of the MBN in the moderate-N treatments (Figure 1). The proportion of applied N acquired by microbes in the low-N treatment was higher (Table 1). There was no interaction effect of total MBN to plant species or N treatment (Figure 1), although the microbial uptake of fertiliser N in mg increased with application rate for both plant treatments ($p \leq 0.001$). Despite no differences in MBN between low

and moderate N application rates, there is a notable reduction in standard error of the moderate-N application rate compared with the low rate for both plant species (Figure 1).

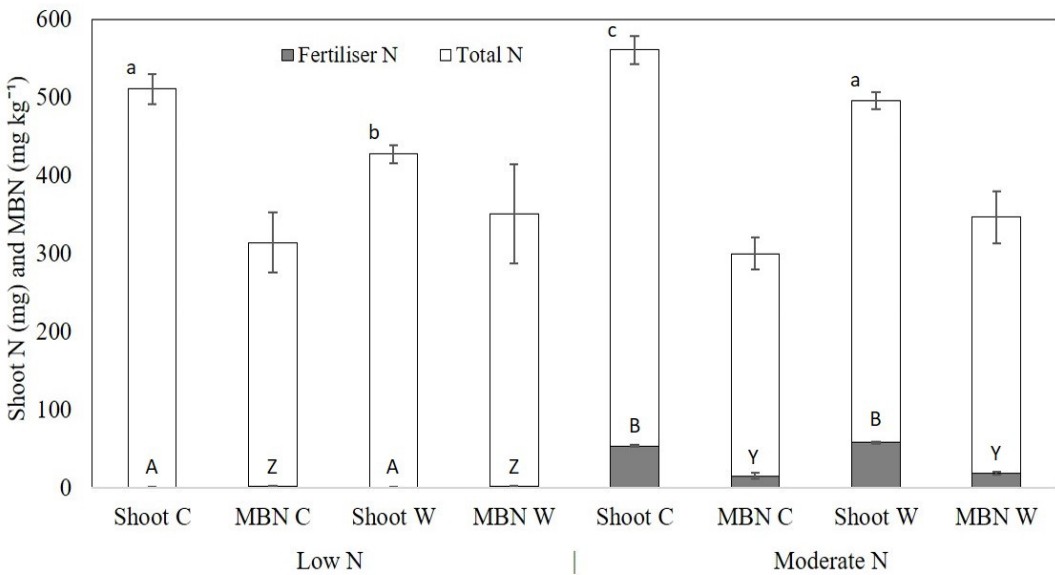

**Figure 1.** Mean N content (mg) of wheat and canola (W and C, respectively) shoots and mean microbial biomass N (MBN, mg kg$^{-1}$) of rhizosphere soil at maturity, showing N sourced from fertiliser (dark grey) and existing soil N pools (white). Error bars represent mean ± standard error. Significant differences, $p \leq 0.05$, indicated by lower case a–c for total plant N, upper case A, B for plant N derived from fertiliser N, and uppercase Z and Y, for MBN derived from fertiliser N. Letters not added where no significant differences occurred.

The mean total N content of plant shoots (mg) at maturity was higher for canola plants (canola = 536, wheat = 462, $p$ = 0.002) and both plant species showed an increase in N content in response to increased N fertilisation (low = 469, moderate = 529, $p$ = 0.006) (Figure 1). There was an increase in the fertiliser N acquired by both species with increased N application ($p \leq 0.001$); however, there was no significant difference between plant species (Figure 1). Whilst the proportion of fertiliser N in wheat shoots was higher than canola (Table 1), this did not equal an increase in total uptake of N due to the difference in overall N content (Figure 1).

Both canola and wheat had similar shoot biomass at anthesis (Table 1), but wheat had higher biomass at maturity (Table 1). The addition of N increased shoot biomass at both anthesis and maturity (Table 1). The concentration of N in plant shoots was higher in wheat at anthesis (Table 1) but higher in canola at maturity (Table 1). Both plant species fell below the threshold for adequate N nutrition as defined by Reuter and Robinson [38] of 1.5% and 1.55%, respectively, for whole shoot N. Thus, in both low- and moderate-N treatments, plants were N-deficient.

### 3.2. Changes in Wheat and Canola Rhizospheres

There was little change in the rhizosphere pH and total mineral N (NO$_3$ and NH$_4$ at 0–10 cm depth) of treatments at anthesis and maturity (Table 1), except for a small increase in total mineral N in the surface soil of canola treatments at maturity (Table 1). The concentration of total N in the rhizosphere soil decreased with increased N application (Table 1). Rhizosphere total N decreased in the moderate-N canola treatment compared with the low N treatment, whilst in both wheat treatments rhizosphere total N remained close to the crop species mean (Table 1).

**Table 1.** Main and treatment effects with statistical significance ($p \leq 0.05$) of plant shoot biomass, shoot N content, applied fertiliser N in plant shoots, applied fertiliser N in MBN, 0–10 cm total mineral N ($NO_3$ and $NH_4$), rhizosphere total N mg kg$^{-1}$, and rhizosphere pH at anthesis and/or maturity. ns denotes no significant main or treatment effects.

| Harvest | | Anthesis | | | | | Maturity | | | | | |
|---|---|---|---|---|---|---|---|---|---|---|---|---|
| | Plant Species | Plant Shoot Biomass (g) | Shoot N Content (%) | 0–10 cm Total Mineral N (mg kg$^{-1}$) | Rhizosphere pH | Plant Shoot Biomass (g) | Applied Fertiliser N in Plant Shoots (%) | Applied Fertiliser N in MBN (%) | 0–10 cm Total Mineral N (mg kg$^{-1}$) | Rhizosphere Total N (mg kg$^{-1}$) | Rhizosphere pH |
| Crop species | Wheat | 28.3 | 1.19 | 1.0 | 6.51 | 47.9 | 6.0 | 6.8 | 1.07 | 1575 | 6.24 |
| | Canola | 28.5 | 1.07 | 1.2 | 6.67 | 44.2 | 4.9 | 6.1 | 1.52 | 1554 | 6.18 |
| | $p =$ | ns | <0.001 | ns | <0.001 | <0.001 | <0.001 | ns | 0.022 | ns | ns |
| N treatment | Low | 27.5 | 1.09 | 1.0 | 6.58 | 43.1 | 0.2 | 8.3 | 1.24 | 1621 | 6.22 |
| | Moderate | 29.3 | 1.17 | 1.1 | 6.66 | 49.0 | 10.7 | 4.6 | 1.35 | 1507 | 6.19 |
| | $p =$ | <0.001 | 0.006 | ns | ns | <0.001 | <0.001 | 0.003 | ns | 0.009 | ns |
| Crop species × N treatment effect | Low wheat | - | - | - | - | - | 0.2 | - | - | 1591 | - |
| | Low canola | - | - | - | - | - | 0.2 | - | - | 1651 | - |
| | Moderate wheat | - | - | - | - | - | 11.7 | - | - | 1558 | - |
| | Moderate canola | - | - | - | - | - | 9.7 | - | - | 1457 | - |
| | $p =$ | ns | ns | ns | ns | ns | <0.001 | ns | ns | 0.035 | ns |
| | l.s.d. | - | - | - | - | - | 0.35 | - | - | 104 | - |

There was no significant difference in the relative abundance of the bacterial phyla present in the rhizosphere of crop species and the unplanted controls (Figure 2a). This remained so throughout taxonomic ranks to the genus level. The relative abundance of fungal phyla (Figure 2b) between treatments was significantly different ($p \leq 0.05$). This significance remained in all taxonomic ranks down to the genus level. There was a large decrease in the presence of Chytridiomycota in the rhizosphere of crop species compared with the unplanted control (Figure 2b). Basidiomycota increased in the planted treatments compared with the unplanted treatments and made up a larger proportion of the fungal community in the wheat rhizosphere compared with the canola rhizosphere (Figure 2b).

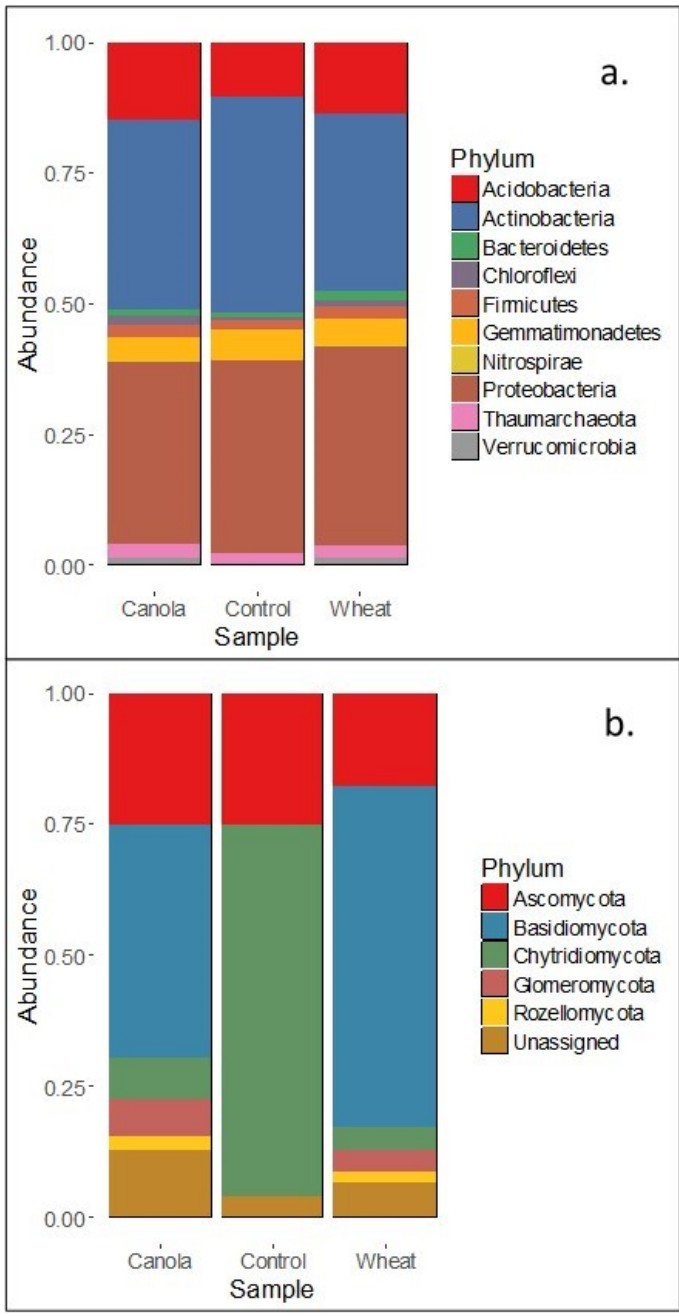

**Figure 2.** Relative abundance of bacterial (**a**) and fungal (**b**) phyla in unplanted controls and the rhizosphere soil of crop species at maturity. Phyla with mean relative abundance <0.001% are not shown.

Visualisation of the microbial data showed that the first two principal coordinates explained the majority of variation in rhizosphere samples (Figure 3a–d). The unplanted controls show separation from the planted treatments on at least one of the first two axis for both the weighted and unweighted UniFrac PCoA, demonstrating plants as major determinants for both fungal and bacterial community composition and structure in the rhizosphere. Figure 3a–c shows a noticeable variation between the planted treatments with plant species having no clear effect on structure. However, the unweighted PCoA for fungal communities did indicate plant species as an explanatory variable for differences in community composition for fungi (Figure 3d). More variation in the microbial communities is explained using weighted UniFrac distances, considering the phylogenetic distance and abundance of OTUs, compared with unweighted UniFrac distances that consider only the species present (approximately 76% and 50%, respectively).

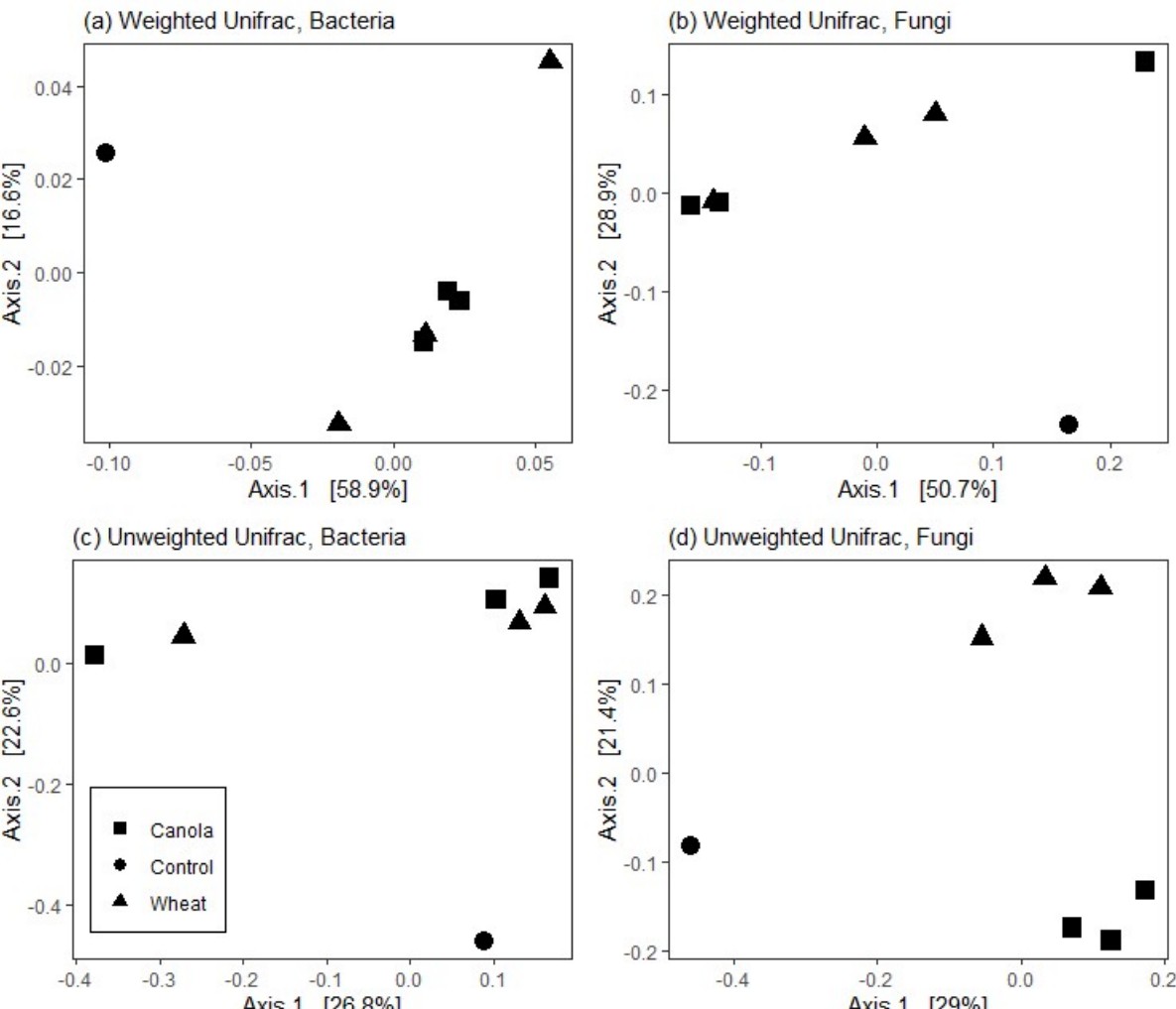

**Figure 3.** Ordination of principal coordinates analysis of weighted (**a**,**b**) and unweighted (**c**,**d**) UniFrac distances between bacterial (**a**,**c**) and fungal (**b**,**d**) communities in the rhizosphere of wheat and canola plants at maturity and no plant controls from the moderate-N treatments. The variation explained by each axis is noted in parentheses.

There was no difference in the rhizosphere enzyme activities measured except for AG (Figure 4). An interaction effect occurred between plant species and N rate ($p = 0.036$), which lead to a decrease in AG activity in the low-N wheat treatment and no significant difference between species in the moderate N treatment. There was also a noticeable increase in the variation of most enzyme activities (as shown by the increase in the standard error) in the planted moderate N treatments compared with the planted low N treatments (Figure 4).

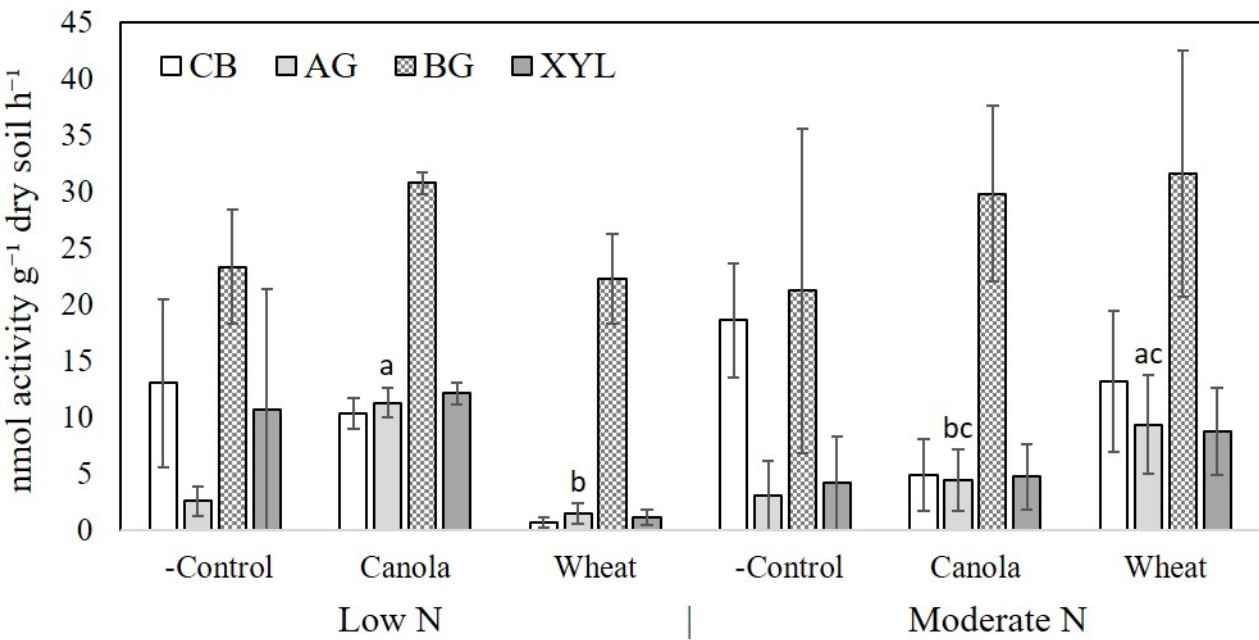

**Figure 4.** Mean extracellular enzyme activity (nmol activity g$^{-1}$ dry soil h$^{-1}$) of cellulose decomposing enzymes (cellobiase, CB; alpha-glucosidase, AG; beta-glucosidase, BG; and beta-xylosidase, XYL) within the rhizosphere of wheat and canola plants and negative unplanted control for low-N and moderate-N treatments at maturity. Error bars represent mean ± standard error. Letters indicate significant differences ($p \leq 0.05$) between treatments. Letters not added to enzymes where no differences between treatments occurred.

## 4. Discussion

The aim of this study was to ascertain whether canola could acquire more fertiliser N than wheat when grown in soil containing C-rich residues by modifying the rhizosphere communities and reducing immobilisation. The results suggest that the competitive ability of wheat and canola to acquire fertiliser N over soil microorganisms is similar despite some differences detected in the rhizosphere microbial populations. The amount of fertiliser N assimilated into plant shoots was similar for both species at both N application rates. Fertiliser N assimilated into the rhizosphere microbial biomass was likewise similar for both plant species at each N rate. The flow of fertiliser N through the soil–plant system therefore appeared to proceed in a comparable manner for both plant species. It is expected that the proportion of fertiliser acquired by plants increased over the growth period. The temporal advantage of plants over microorganisms in attaining N is due to differences in life/death cycle rates, and the capacity of plants to further accumulate mineralised nutrients [8]. In our experiment, the amount of N fertiliser in plant shoots and the rhizosphere microbial biomass at maturity was more dependent on the N application rate than on plant species.

The increased amount of existing soil N in canola shoots and rhizosphere microbial biomass suggests that canola may better attain N from soil. The total mass of shoot N not originating from fertiliser was consistently higher in canola shoots compared with wheat (mean difference ~76 mg). Whilst there was no statistically significant difference between the rhizosphere MBN attained from soil, the mean for canola was lower than that of wheat (mean difference ~39 mg kg$^{-1}$), and the variation decreased with moderate N fertiliser application (reduced standard error with increased N, Figure 1). This reduced variation suggests that the competition for N may have been decreasing with increased N availability from fertiliser application. Differences in rhizosphere MBN between wheat and canola may be detectable when adequate N is present or with increased replication of treatments. Given that both plant species were deficient in total shoot N at maturity, it appears that the whole system was N-stressed throughout the experiment. Therefore, the lower mean MBN obtained from soil in the canola rhizosphere may have partially contributed to the increase

in canola shoot N. Whilst this mechanism is speculative, the plant shoot N differences between canola and wheat show that canola was consistently better at obtaining soil N.

The results of Ryan et al. [12] suggest that the composition of brassica tissues and root exudates select soil microbial communities which facilitate increased mineralisation of N compared with non-brassica crops. Our results showed that the rhizosphere fungal communities of wheat and canola differed at plant maturity and that the total mineral N in the surface 0–10 cm was higher for canola than wheat (albeit by a small margin). Increased mineralisation of soil N throughout the growth period may have facilitated the increase in soil N assimilation seen in canola shoots. Whilst the relative size of the fungal and bacterial populations was not determined, the lack of difference between the rhizosphere bacterial communities suggests that the fungal communities present made an important contribution to the increase in soil N mineralisation in the canola treatments.

It is well documented that bacterial rhizosphere communities associated with plant species differ and can even vary between cultivars [39]. The specificity of fungal communities to plant species is not as reliably reproduced in the literature [40]. However, individual species such as mycorrhizae are closely associated to specific plant species [41]. We found differences in the relative abundance of fungal phyla present in wheat and canola rhizospheres but no difference in bacterial phyla. Selective pressure for decomposers to proliferate in the surface 10 cm of soil in our treatments would have been high due to the incorporation of C-rich wheat residues. Bacteria may have been outcompeted for soil resources as the most abundant C source was exploited by fungi. Fungi may also suppress bacteria via the production of antibacterial compounds [42] which may have adversely affected bacterial growth near hyphae. Basidiomycetes and actinomycetes made up approximately ~69% of the rhizosphere fungal communities of canola (44 and 25%, respectively) and ~82% of wheat plants (64 and 18%, respectively) and microbial degradation of lignin is largely confined to these two phyla [43]. Lignin content of wheat straw is variable depending on the cultivar and growth conditions. Summerell and Burgess [44] examined wheat straw decomposition in the field and the laboratory, and reported initial lignin contents of wheat straw ranging from 5.2–11.2% dry matter, which increased as the wheat straw decomposed. As such, elevated lignin content of wheat straw may have enabled basidiomycetes and actinomycetes to flourish, whilst the suppression of bacteria by fungi reduced their proliferation.

The importance of soil microbial communities to N mineralisation is contentious as this process is facilitated by most soil microbial species. Blagodatskaya and Kuzyakov [18] reasoned that differences in the taxonomic composition of the microbial population needs to be linked to differences in the functional activity of the community present to determine the capacity for differential substrate decomposition, including SOM mineralisation. We did not see any differences in the enzyme activities measured in soil except for AG, which degrades disaccharides into glucose. Increased activity of extracellular cellulose and lignin degrading enzymes have been linked to increased SOM decomposition [45,46]. We did not observe any difference in BG or CB activity amongst treatments, which are associated with cellulose decomposition, and did not measure ligninolytic activity of rhizosphere soil at maturity.

The fungal relative abundance data suggests that lignin decomposers were dominant in the community at plant maturity. The proportion of the fungal community which was lignin-degrading was larger in the wheat treatments as was the predicted activity of saprotrophs (data not shown). These facts do not support the premise that the canola microbial community was responsible for increased mineralisation. However, the description of the microbial communities and enzyme activities provided here represent the rhizosphere conditions and microbes present at maturity and not throughout plant growth, which would have facilitated increased soil N availability to canola. It is expected that microbial communities and enzyme activities changed over time with the successive generations of soil microorganisms in response to the changing rhizosphere conditions during plant growth. As such, whilst the singular sampling date reasonably reports the proportion of

fertiliser N in the plant and microbial biomass, it does not allow for the determination of the microbial communities that resulted in the differences in soil N assimilation by plants.

Some of the difference in N content between canola and wheat may be due to the differences in reproductive development and duration of the total growing period. As mentioned previously, plants attain a greater proportion of available N than microorganisms with increased time. Canola has an indeterminate life cycle resulting in a period of prolonged flowering before senescence [47]. Conversely, wheat is determinate with highly synchronous flowering and faster senescence. The indeterminate habit of canola means that it still had living shoot and root tissue at the sampling time of maturity for wheat in this experiment. Therefore, it is likely that some of the difference in total N content of plant shoots can be attributed to the prolonged growth period of canola. The extent to which this affected the results, however, is unclear as no difference was observed in fertiliser N uptake, which is also expected to increase if this factor was important. In field conditions, canola growth will be largely terminated when windrowed or desiccated with herbicides, with any increased N-uptake occurring through this mechanism being retained in the system as plant residues.

## 5. Conclusions

The partitioning of fertiliser N into plants and microorganisms appears to occur similarly in wheat and canola in the conditions observed throughout this experiment. Whilst the conditions used are not fully representative of those in the field, the results provide some insight into the source of N acquired by wheat and canola when immobilisation pressure is high. No difference in fertiliser uptake was evident between the canola and wheat plants, and immobilisation of N fertiliser did not differ in plant rhizospheres. Canola obtained an elevated amount of N from existing soil sources. Reduced mean MBN acquired from soil in the canola rhizosphere may account for some of the difference between the two plant species. Whilst no significant difference in rhizosphere MBN between wheat and canola treatments was observed, insufficient N supply to the system is likely the reason. Differences in the enzyme activities and relative abundance of soil microorganisms were present, although they did not appear to correlate with increased mineralisation of soil N by canola. This discrepancy is possibly due to the sampling date not correlating with the actual mineralisation events that lead to increased soil N uptake by canola.

Future investigations into the effects of plant species on soil N cycling would benefit from considering the influence of microbial succession in the rhizosphere, and how plant development affects nutrient partitioning. Whilst shorter studies assessing differences before reproductive growth of plants are valuable, they lose a crucial period of increased plant nutrient demand which impacts rhizosphere interactions. As such, including a larger number of sampling intervals would provide insights into these interactions over time. Additionally, including treatments with adequate N for plant growth would allow clearer quantification of the ability of plants to influence soil microorganisms and their requirement for N.

**Author Contributions:** Conceptualization, J.R.H., J.A.K., N.N. and B.A.R.; methodology, B.A.R., N.N., J.R.H., C.T. and C.C.; validation, J.R.H., C.T. and C.C.; formal analysis, B.A.R., N.N. and C.C.; investigation, B.A.R., N.N. and C.C.; resources, J.R.H., C.T. and C.C.; data curation, B.A.R., N.N. and C.C.; writing—original draft preparation, B.A.R.; writing—review and editing, B.A.R., N.N., C.C., J.R.H., J.A.K. and C.T.; visualization, B.A.R. and C.C.; supervision, J.R.H. and C.T.; project administration, B.A.R., N.N. and J.R.H.; funding acquisition, J.R.H. and C.T. All authors have read and agreed to the published version of the manuscript.

**Funding:** This research was funded by La Trobe University via its Honours and Ph.D. scholarship programs.

**Data Availability Statement:** Not applicable.

**Conflicts of Interest:** The authors declare no conflict of interest.

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
