# Peer review of "Nitrogen Fertiliser Immobilisation and Uptake in the Rhizospheres of Wheat and Canola"

_agronomy, doi:10.3390/agronomy11122507_

Round 1

Reviewer 1 Report

All my comments, suggestions, questions and corrections are in sticky notes in the attached pdf file of the manuscript.

Reviewer 2 Report

The details of review are as follows:

  1. Authors must check the English expression, because it is really, really very very difficult for readers to understand what your manuscript wanted to express. Language expression should be concise. The phrases which are not important and redundant should be deleted, such as present, throughout the experiment …… the comparison between A and B should be expressed clearly.

Introduction

  1. L67: O'Sullivan, Duncan, Whisson, Treble, Roper and Peoples ….: not all the authors should be listed.

Materials and Methods

  1. L141: Senesced leaves were collected during the growth period as they detached from the…: what was the aim of collecting leaves? When collected? Which index were measured? No data was demonstrated in the paper.

Results

  1. L248-249: Whilst fertiliser N made up more of the MBN in the moderate-N treatments (Figure 1), the proportion of applied N acquired by microbes in the low-N treatment was higher (Table 1). Who compares with whom? between HN and LN or others? The treatment in the text is moderate-N treatment, but the treatment in the figures is high N treatment, which is correct? Please unify them. Are these two sentences contradictory?
  2. L251: although the microbial uptake of fertiliser N increased with application rate for both plant species……: the meaning of the sentence is contradictory to the sentence of “the proportion of applied N acquired by microbes in the low-N treatment was higher (Table 1).”?
  3. L266: however? Please check the use of the word, Punctuation should be added before and after the word. Please check throughout the paper.

Table 1: Make this table in a page with horizontal distribution, the content in each row should be in the same row.  Why are some words in bold type and the other words not? Some are italicized, some are not? Unified according to the required format. Why he indexes of “Applied fertiliser N in plant shoots (%) Applied fertiliser N in MBN (%) Rhizosphere total N (mg kg-1) “ were not measured at anthesis?

  1. L287-289: There was little change in the rhizosphere pH and total mineral N (NO3 and NH4 at 0-10cm depth) of treatments at anthesis and maturity (Table 1), although canola had more total mineral N in the surface 0-10cm of soil at maturity. The meaning of the two sentences were contradict to each other, please clarify them.
  2. L292: “of wheat remained close to the crop species mean (Table 1)” please reorganize the English.
  3. L298,299 : Chytridiomycota OTUs? Basidiomycete OTUs? OUT or Phyla?
  4. Basidiomycete were higher in crop treatments than in control, the author did not show it.

11 L310: the microbial communities’ ?? what is the symbol of ’?

12: L314-315: There was no difference in the enzyme activities of rhizosphere and unplanted soil measured except for AG (Figure 4) authors did not perform the Analysis of Variance.

13: L319: (Figure 5)? Where is Figure 5?

Discussion

  1. L352: the variation decreased with increasing N fertiliser application (reduced standard error with increased N, Figure 1) why is the variation? ??

15: L393-416: what is the aim of the paragraph? Do you mean the relationship between microbial community and N mineralization? What relationship were between them? Which microorganisms can mineralize N? in this study, which phyla played a key role in the N mineralization?

16: L404: Additionally, the proportion of the fungal community which was lignin degrading was larger in the wheat treatments as was the proportion of saprotrophs (data not shown) ??? who was larger than who? Please clarify it.

17: L413-415: As such, whilst the singular sampling date reasonably reports the proportion of fertiliser N in the plant and microbial biomass, it does not allow for the determination of temporal changes in the microbial community composition and activity that resulted in the differences in soil N assimilation. Why did not the authors measure the microbial community at anthesis?

Figures

Figure 1: Analysis of Variance should be added to the figure and other figures.

Round 2

Reviewer 2 Report

accepted